# Psychosocial Outcomes in Parents of Children with Acute Lymphoblastic Leukaemia in Australia and New Zealand Through and Beyond Treatment

**DOI:** 10.3390/cancers17071238

**Published:** 2025-04-06

**Authors:** Clare Parker, Clarissa E. Schilstra, Karen McCleary, Michelle Martin, Toby N. Trahair, Rishi S. Kotecha, Shanti Ramachandran, Ruellyn Cockcroft, Rachel Conyers, Siobhan Cross, Luciano Dalla-Pozza, Peter Downie, Tamas Revesz, Michael Osborn, Glenn M. Marshall, Claire E. Wakefield, Marion K. Mateos, Joanna E. Fardell

**Affiliations:** 1Children’s Cancer Centre, Monash Children’s Hospital, Clayton, VIC 3168, Australia; clare.parker@monashhealth.org (C.P.);; 2Behavioural Sciences Unit, Kids Cancer Centre, Sydney Children’s Hospital, Randwick, NSW 2031, Australia; 3Discipline of Paediatrics, School of Clinical Medicine, UNSW Sydney, Kensington, NSW 2033, Australia; 4Kids Cancer Centre, Sydney Children’s Hospital, Randwick, NSW 2031, Australia; 5Children’s Cancer Institute, Lowy Cancer Research Centre, UNSW Sydney, Kensington, NSW 2033, Australia; 6Department of Clinical Haematology, Oncology, Blood and Marrow Transplantation, Perth Children’s Hospital, Nedlands, WA 6009, Australia; 7Leukaemia Translational Research Laboratory, WA Kids Cancer Centre, The Kids Research Institute Australia, Nedlands, WA 6009, Australia; 8Curtin Medical School, Curtin University, Bentley, WA 6102, Australia; 9Medical School, University of Western Australia, Crawley, WA 6009, Australia; 10Starship Children’s Hospital, Auckland 1023, New Zealand; 11Royal Children’s Hospital, Parkville, VIC 3052, Australia; 12Department of Paediatrics, Melbourne University, Parkville, VIC 3052, Australia; 13Murdoch Children’s Research Institute, Parkville, VIC 3052, Australia; 14Children’s Haematology Oncology Centre, Christchurch Hospital, Christchurch 4710, New Zealand; 15Cancer Centre for Children, The Children’s Hospital at Westmead, Westmead, NSW 2145, Australia; 16Women’s and Children’s Hospital, North Adelaide, SA 5006, Australia; 17Adelaide Medical School, The University of Adelaide, Adelaide, SA 5005, Australia

**Keywords:** childhood acute lymphoblastic leukaemia, parental distress, health-related quality of life, psychosocial

## Abstract

Over recent decades, outcomes for children with acute lymphoblastic leukaemia (ALL) have significantly improved. However, parents often experience distress and psychological challenges throughout their child’s treatment. Our study explores the levels of distress, anxiety, depression, anger and the need for help among parents in Australia and New Zealand who have a child diagnosed with ALL. We enrolled parents prospectively and had them complete surveys at pre-defined time points to track their psychological experiences across the patient journey.

## 1. Introduction

Significant advances in treating paediatric patients with acute lymphoblastic leukaemia (ALL) have led to a five-year overall survival rate of ≥ 90% in high and middle-income countries [1,2]. Therefore, there is growing emphasis on reducing morbidity associated with this condition, which includes a need to explore wellbeing for the entire family unit. Among these efforts, addressing the psychological and emotional outcomes of parents and caregivers of children with ALL is a key area of focus.

Parents and caregivers of children with cancer, including ALL, experience heightened levels of distress compared to the general population [3,4]. The treatment process often involves prolonged hospitalizations and invasive procedures and there is a high burden on parents who often deliver daily medications and monitor side effects. Parental function plays a pivotal role in the child’s experience of cancer treatment, and the bidirectional relationship between parent and child wellbeing has been well described [5,6,7]. The distress and emotional burden on parents has been shown to negatively impact the quality of life for both parents themselves as well as children [8,9,10]. In contrast, effective parental coping strategies and emotional resilience have been shown to positively influence the child’s adaptation to their illness and overall health-related quality of life (HRQoL) [11].

The previous literature has documented the emotional response of parents and carers to having a child diagnosed with leukaemia and explored the HRQoL of parents in a variety of geographical and socio-economic settings [12,13,14,15,16,17,18,19,20]. However, most existing leukaemia-specific studies are either qualitative or cross-sectional, retrospective and involve small population sizes, resulting in a lack of longitudinal data [8]. Furthermore, there is a lack of standardization regarding the time points at which distress is measured and the methods used to report the prevalence, severity and duration of symptoms [21]. Notably, the contemporary psychological experiences of Australian and New Zealand parents and caregivers remain underexplored; prior studies are now over 20 years old and included fewer than 40 participants [22,23].

Parents’ and caregivers’ experiences are influenced by a range of positive and negative factors. Factors such as higher income, better baseline mental health, lower levels of care burden with less daily care time and having more co-caregivers such as extended family and social supports have been shown to positively impact parents’ emotional wellbeing [18,24]. For example, qualitative data from parents of children diagnosed with cancer suggest that the experience can strengthen family cohesion and draw family members closer together [11]. Conversely, social isolation, increased sleep disruptions, negative attachment style and female gender have been previously associated with negative parental HRQoL [25,26,27]. For example, data from a cross-sectional survey of 44 caregivers demonstrated significantly higher anxiety scores (measured by the Beck Anxiety Inventory) in women compared to men [28].

Psychosocial assessment is a standard of care in paediatric oncology practice, and existing evidence supports the use of tools such as the Psychosocial Assessment Tool (PAT) and Distress Thermometer (DT) for early identification of risks and resiliencies [29]. Additionally, a large body of evidence supports the use of the Patient-Reported Outcome Measurement Information System (PROMIS) in paediatric oncology trials, with strong evidence for validity and responsiveness [30].

Our study uses a prospective, longitudinal approach to assess parental distress, anxiety, depression, anger and the need for help at multiple time points during diagnosis and treatment. We have previously shown that prospectively monitoring the incidence of treatment-related toxicities and the longitudinal impacts on HRQoL throughout ALL therapy is feasible in the Australian/New Zealand setting [31]. This work found that parents’ emotional wellbeing was poorest in the first 6 months post-diagnosis, with anxiety being the most highly rated concern. In this study, we expand on our previous work with a longer duration of follow-up (5 years) and a larger parent cohort (117 participants). The primary aim was to describe the psychological experiences of Australian and New Zealand parents of children diagnosed with ALL, focusing on the prevalence and time course of psychological comorbidities including distress, anxiety, depression, anger and the need for help. Secondary aims included (1) identifying potential risk factors for worsened parental psychosocial outcomes, such as time since diagnosis, child demographics (age and sex), socio-economic status and disease-related variables (risk stratification and presence of toxicity), and (2) evaluating the correlation between the ET tool responses and PROMIS responses.

## 2. Materials and Methods

### 2.1. Participants

The *Acute Lymphoblastic Leukaemia Subtypes and Side Effects from Treatment* (ASSET) study is a prospective registry study conducted across nine sites in Australia and New Zealand, designed to identify ALL patients at increased risk of treatment-related toxicity and capture whole-of-life impacts from ALL and its treatment. Patients and their parents were identified by their treating clinicians and invited to participate by a research coordinator. Patients were recruited within 90 days of commencing chemotherapy. The ASSET study opened to recruitment in January 2016, and the HRQoL opened to recruitment in October 2018 at eight of the nine sites. Here, we report on data obtained from the HRQoL sub-study between October 2018 and November 2022. Eligibility criteria for the HRQoL sub-study included parents/guardians of children aged ≤ 18 years old with newly diagnosed ALL or mixed phenotype acute leukaemia (MPAL). The study was approved by the HNE HREC (2019/ETH00693) and conducted according to the Australian National Statement on Ethical Conduct in Human Research [32]. Informed voluntary consent was obtained from participants or their parents and/or legal guardians.

### 2.2. Measures

Following consent, eligible parents were sent an invitation email to complete their study questionnaire via Research Electronic Data Capture (REDCap). Families nominated one parent or carer to complete all assessments, which were completed online using any internet-enabled device. Surveys were distributed at set time points using two tools to measure parental psychosocial wellbeing. Parents were invited to complete surveys every two months during the first two years of their child’s treatment, followed by surveys every six months for the next three years. The central ASSET study team managed survey distribution. We extracted demographics and clinical factors including risk stratification and treatment-related toxicity from the larger ASSET database.

The Emotion Thermometer (ET) tool is an adaptation of the distress thermometer, which was developed and validated for the evaluation of distress in cancer [33,34,35]. The ET assesses distress, anxiety, depression, anger and the need for help, and is scored on a 0–10 scale [36]. The ET tool has been validated in an Australian oncology population [37]. The ET tool is a single-item questionnaire used for screening.

Participants also completed Patient-Reported Outcome Measurement Information System (PROMIS) questionnaires for anxiety, depression and anger [38,39]. The PROMIS surveys consisted of 29 anxiety statements, 28 depression statements and 22 anger statements. Participants rated their response to each statement on a scale from always, often, sometimes, rarely to never. PROMIS questionnaires have been validated in the Australian adult population [40]. As they consist of multiple questions, PROMIS questionnaires are a more thorough assessment compared to the ET screening tool.

In summary, we used five ET measures (distress, anxiety, depression, anger and the need for help) and three PROMIS measures (anxiety, depression and anger). To minimise the burden of participation for parents, PROMIS anger questionnaires were removed from the study in December 2021, and anger was subsequently only measured using the ET tool.

### 2.3. Statistical Analysis

We conducted all data analysis using IBM SPSS Statistics Version 29.0.2.0. To measure trends over time, we grouped survey responses based on time since diagnosis (0–6 months, 6–12 months, 12–18 months, 18–24 months, 2–3 years and ≥3 years). We conducted single and multiple variable mixed model analyses to analyse for associations with increased parental distress responses.

We classified ET tool scores greater than or equal to 4 as clinically significant and warranting further follow-up according to previously published clinical cut-offs [41,42,43].

PROMIS T-scores were assessed based on the developers’ instructions. T-scores 0.5–1 standard deviation above the mean T-score were classified as mildly elevated. Scores between 1 and 2 standard deviations were classified as moderately elevated. Scores 2 or more standard deviations above the mean were classified as severely elevated [38,44,45].

We used Pearson correlations to evaluate the relationship between mean ET tool responses and mean PROMIS questionnaire responses for anxiety, depression and anger. We calculated Kappa statistics to determine the level of agreement between clinically elevated responses measured by the ET tool and PROMIS questionnaires. We interpreted the Kappa statistic of < 0.0 to demonstrate no agreement, 0–0.20 as slight agreement, 0.21–0.40 as fair agreement, 0.41–0.60 as moderate agreement, 0.61–0.81 as substantial agreement and 0.81–1.0 as almost perfect agreement.

Regionality and socioeconomic data were determined based on patient postcode. For Australian participants, the Accessibility/Remoteness Index (ARIA) was used to classify parents as residing in metropolitan, inner regional, outer regional or remote areas [46]. Postcode was also used to determine the parent Socio-economic Index for Areas (SEIFA) score, which was used for socioeconomic classification [47]. A lower SEIFA score indicates a higher level of socio-economic disadvantage.

### 2.4. Psychological Safeguards

For parents indicating a clinically elevated need for help, they received a follow-up phone call from a research psychologist and were offered information resources and information about local supportive services via e-mail. Parents were also given the opportunity to have their concerns raised with their child’s local treating team, or the opportunity to discuss their need for help with a study psychologist via telephone conversation.

## 3. Results

Of the 297 patients enrolled in the ASSET registry between October 2018 and November 2022, 119 parents (40%) participated in the opt-in HRQoL sub-study. Two patients were excluded due to being over 18 years old. From 117 eligible parents, 327 survey responses were collected (Table A1), ranging from 1 to 8 responses per participant, with a median of 2 surveys per parent. One-third (34.2%) of parents completed only one survey. The survey responses spanned from 0 to 62 months since diagnosis (Figure A1). There were 67 responses recorded from 0–6 months post-diagnosis, 46 from 6–12 months, 48 from 12–18 months, 48 from 18–24 months, 63 for 2–3 years and 55 for >3 years. There were a few missing responses (<2% for each measure).

### 3.1. Demographics 

The demographics of study participants are shown in Table 1. The median age of children at ALL diagnosis was 7 years, ranging from 6 months to 18 years, with the majority (n = 85, 73%) being under 10 years old. Most children with ALL (n = 71, 61%) were male. Approximately two-thirds of children with ALL (n = 75, 64%) were treated on Children’s Oncology Group (COG) treatment protocols, while 42 (36%) received treatment on Berlin–Frankfurt–Münster (BFM) protocols. Children’s diagnoses were similarly distributed across risk categories, with 41 (35%) low/standard risk, 35 (30%) medium risk and 41 (35%) high risk. Most families (n = 90, 77%) resided in metropolitan areas, which is broadly representative of urban–rural distributions in Australia (where 73% of the total population reside in metropolitan areas [48]) and New Zealand (where over 80% of the population reside in urban areas [49]). The median SEIFA centile was 60.5 (range 5–100), with about half of the participants (n = 59, 50.5%) falling in the 60th centile or above. During the data collection period, four children (3%) experienced disease relapse and two children (2%) died.

### 3.2. Trends over Time

The proportion of parents demonstrating clinically elevated emotional scores reduced over time, as shown in Table 2. Across the eight measures (five ET and three PROMIS measures), fewer participants reported clinically elevated scores as time since diagnosis increased. Within the first months following diagnosis, 40.3% of parents reported four or more clinically elevated measures, which decreased to 23.6% beyond three years. Only one-third (33.6%) of parents reported no clinically elevated measures at any time point.

ET tool responses for parent QOL measures peaked within the first 6 months and gradually reduced over time (Figure 1A). The mean score for distress and anxiety was above the clinical cut-off within the first 6 months, highlighting the significant emotional response during this time point.

As measured by PROMIS questionnaires, anxiety and depression levels were highest within the first six months, followed by a gradual reduction over time (Figure 1B). In contrast, anger remained relatively stable throughout the study period, showing no significant temporal variation.

### 3.3. Emotion Thermometer Responses

As measured by the ET tool, anxiety was the most frequently elevated measure. Anxiety scores were highest within the first 6 months post-diagnosis, with a mean ET score of 5.0 and 64% of parents scoring ≥ 4 (Figure 2A). By 18–24 months, the mean ET anxiety score decreased to 3.5, although nearly half (48%) of responses remained above the clinical cut-off. From 2 years post-diagnosis and beyond, the mean anxiety score rose slightly to 4.0, with a similar proportion of responses scoring above the clinical cut-off (51% between 2 and 3 years, 47% beyond 3 years).

Mean ET scores for depression remained relatively stable over time (range 2.19–2.93 across periods), though the proportion of parents reporting clinically significant depression decreased from 40% within the first 6 months post-diagnosis to 26% beyond 3 years (Figure 2B).

Mean ET anger scores remained consistent over time, ranging from 2.11 to 2.67 across different time periods. The proportion of parents reporting clinically elevated anger (score ≥ 4) varied between 25% and 31% without a significant peak at any specific time period (Figure 2C).

Distress scores were highest within the first 6 months of diagnosis, with 55% of parents/carers reporting scores above the clinical cut-off (Figure 2D). A gradual reduction in distress scores was observed over time, with 28% of parents reporting clinically elevated distress more than 3 years post-diagnosis.

The need for help scores peaked within the first 6 months post-diagnosis, with 39% of parents/carers reporting scores above the clinical cut-off (Figure 2E). This proportion decreased over time, reaching 19% for those who were more than 3 years post-diagnosis, reflected by a gradual decline in mean scores to 1.69 after 3 years.

### 3.4. PROMIS Questionnaire Responses

Anxiety was also the most commonly elevated measure on PROMIS questionnaire responses. In the first six months from diagnosis, the mean PROMIS T-score was 56.3, with 34% in the moderate to severe range (Figure 3A). In difference to ET tool responses, there was a larger proportion of responses indicating clinically elevated anxiety between 6 and 12 months from diagnosis. There was a secondary peak 18–24 months from diagnosis, reflecting a time approaching the end of treatment in most ALL protocols. The mean PROMIS anxiety T-score decreased to 50.8 between 2 and 3 years.

PROMIS depression scores were highest within the first 6 months of diagnosis, with a mean of 53.3 and 21% in the moderate to severe range (Figure 3B). By 3 years post-diagnosis, the mean score dropped to 49.0, with only 7% of parents scoring in the moderate to severe range.

Mean PROMIS anger T-scores ranged from 49.2 to 51.6 across time blocks. The highest proportion of anger scores occurred between 6 and 12 and 18 and 24 months post-diagnosis, with over 27% indicating moderate to severe anger (Figure 3C). This contrasts with 15% within the first 6 months post-diagnosis and less than 13% from 2 years onwards.

### 3.5. Risk Factor Analysis

As time from diagnosis increased, and adjusting for other variables in the model, scores significantly decreased for PROMIS anxiety (estimate −0.18, 95% CI −0.28 to −0.086, *p*-value <0.001), PROMIS depression (estimate −0.13, 95% CI −0.22 to −0.0036, *p*-value = 0.006) and ET distress (estimate −0.073, 95% CI −0.13 to −0.015, *p*-value = 0.013) (Table 3). Using a single variable analysis, increasing time was associated with reduced need for help scores (estimate −0.018, 95% CI −0.036 to 0.00, *p*-value = 0.045), though this association was not statistically significant on multiple variable mixed model analysis.

In this dataset, no statistically significant associations were found between the measured emotional responses and other independent variables including the child’s age at diagnosis, the child’s sex, ALL risk stratification, or the presence of toxicity (Table 3). We found a statistically significant association between the ET anger score and the SEIFA index, whereby a higher SEIFA index score was associated with a lower ET anger score (estimate −0.02, 95% CI −0.03 to −0.001, *p*-value = 0.038). In our single variable mixed model analysis, we noted a statistically significant association with the study group protocol (COG or BFM) the child was treated on and ET anger score (mean difference estimate −0.76 for children treated on BFM study group protocols, 95% CI −1.49 to −0.04, *p*-value = 0.040); however, this did not remain significant when adjusting for other variables in the multiple variable mixed model analysis. 

### 3.6. Correlation Between the Emotion Thermometer Tool and PROMIS Questionnaires

There was a moderately strong correlation between anxiety and depression scores when measured by the ET tool compared to PROMIS measures (Pearson correlation 0.44 and 0.34, respectively), whereas the correlation between anger scores was less strong (Table 4). However, the correlation between measures was statistically significant (*p*-value <0.05) for all measures.

There was statistically significant agreement between anxiety and depression scores above clinical cut-off when measured by both the ET tool and PROMIS questionnaires (Table 5). Anger demonstrated lower level of agreement that was not statistically significant.

## 4. Discussion

Our results describe the psychological morbidity experienced by parents of children diagnosed with ALL in the Australian and New Zealand context. Anxiety emerged as the predominant psychological burden, with significant levels observed in over 50% of ET responses and moderate to severe symptoms observed in 30% of PROMIS questionnaires. Substantial rates of other psychological symptoms were also demonstrated: depression (29% ET, 15% PROMIS moderate–severe), anger (27% ET, 19% PROMIS moderate–severe), distress (39% ET) and the need for help (24% ET). Temporal patterns of emotional distress were observed, with anxiety and depression peaking within the first six months post-diagnosis, followed by a gradual decline over time. Conversely, parental anger, which has been less well explored in the existing literature, remained relatively stable throughout and did not exhibit an early peak. Despite the overall reduction in average distress, anxiety, depression and the need for help scores over time, a significant minority of parents continued to experience psychological challenges beyond two years post-diagnosis. Additionally, a second peak in anxiety was identified around the two-year mark, which is when ALL treatment often approaches a conclusion. Temporal proximity to diagnosis emerged as the only risk factor for increased psychological morbidity in this cohort. For the psychological symptoms measured by both the ET and PROMIS questionnaires, strong correlations were observed between anxiety and depression responses on the different tools, whereas anger exhibited a more moderate but statistically significant correlation. Furthermore, there was significant agreement between clinically elevated measures across both assessment instruments.

Our study is the largest prospective investigation of parental psychological wellbeing in paediatric ALL within Australia and New Zealand, providing robust evidence of the significant emotional comorbidities in this population. The results of the ASSET parental HRQoL sub-study build on what has been shown in the literature in other populations, with high levels of parental anxiety, depression and distress. Population-level data from Australia [50] and New Zealand [51] suggest prevalence of mental health disorders is around 20% among adults. In contrast, our results revealed a higher prevalence of clinically significant anxiety and depression, underscoring the heightened psychological burden in this group compared to the general Australian and New Zealand populations. Prior studies, noting that many group together cancer types, have similarly demonstrated a high prevalence of negative psychological wellbeing among parents; for example, in a cross-sectional study, 41% of 518 surveyed caregivers reported always/often experiencing depressive or anxiety symptoms [52].

Our study is unique due to its prospective, longitudinal design, which included up to five years of follow-up for parents. This design enabled the identification of chronological trends in emotional responses and allowed for the analysis of potential demographic or disease-related correlations with psychosocial outcomes. The temporal patterns of emotional distress observed in this study build upon the existing literature, which has often grouped all cancer types together, providing new insights specific to this population. For example, in a previous longitudinal investigation of the trajectory of caregiver psychological symptoms of anxiety, depression and distress, caregivers endorsed more elevated symptoms at diagnosis compared to 6 months later [53,54]. The second peak of anxiety observed towards the end of treatment likely reflects the new emotional challenges arising during this transition. Previous studies have identified the unmet needs faced by families as they prepare for post-treatment challenges, which has been hypothesised to contribute to increased distress at this time point [55,56].

The association between increased time since diagnosis and reduced anxiety, depression and distress scores suggests a degree of emotional adjustment over time. The intensity of ALL treatment in the first 6 months is also a likely contributing factor to high levels of psychological distress during that time. This interpretation is consistent with findings of previous prospective studies in which anxiety symptoms declined over the first year of treatment but remained elevated compared to parents who did not have a child diagnosed with cancer [57,58]. The persistent psychological challenges faced by a subset of parents highlight the importance of early and broad psychological screening at the time of diagnosis to identify parents experiencing significant distress [4,59]. Such screening could enable the timely identification of parents at higher risk of distress, facilitating targeted interventions and appropriate resource allocation to support the entire family’s psychological wellbeing.

We also assessed parental anger in our population, a novel area of investigation that has been largely unexplored in previous studies on parental emotional wellbeing. Anger scores measured using the ET screening tool showed a significant correlation with those measured by the PROMIS questionnaire (Pearson correlation coefficient: 0.17, 95% CI: 0.05–0.30, *p* = 0.008). However, there was limited agreement between clinically significant anger as identified by the ET tool and the PROMIS questionnaire (Kappa statistic: 0.09, *p* = 0.060). The ET tool, as a screening measure, appears to provide a quick assessment of anger but may lack the nuance of the PROMIS questionnaire. This could explain the limited agreement in identifying clinically significant anger. The PROMIS tool is designed to evaluate emotions across multiple dimensions, allowing for greater precision in identifying thresholds of clinical significance. Anger may differ from depression and anxiety in how it manifests and is reported, potentially influencing the utility of a screening tool like ET. Anger might be more situational or reactive, whereas depression and anxiety often reflect more pervasive emotional states. This variability could reduce the sensitivity and specificity of simpler tools in detecting anger compared to other emotions.

Our study identified an association between lower socio-economic status, as measured by the SEIFA index, and higher ET anger scores. This finding may highlight the systemic barriers faced by different patient cohorts when navigating the healthcare system. In a single variable analysis, we observed an association between the treatment protocol the child received and ET anger scores. However, the small sample size of patients treated with BFM protocols, coupled with the lack of a consistent association in the multivariable analysis, suggests that this finding may be attributable to limitations in sample size.

### 4.1. Strengths and Limitations

This is the first bi-national study investigating the longitudinal emotional wellbeing of Australian and New Zealand parents and carers of children with ALL. We included both screening measures (the ET tool) and a more comprehensive evaluation (PROMIS questionnaire) of emotional wellbeing and followed parents for more than 3 years post-diagnosis. However, there are some limitations worth noting. Longitudinal studies are affected by retention and subsequent representation of the population. Of the eligible ASSET population, whilst 40% participated in the HRQoL sub-study, there was a high rate of attrition with one-third of the participants not completing more than one survey. Parents with children in more critical health conditions may have lacked the time or ability to participate fully, as their energy may have been directed towards care for their unwell child, suggesting that our results may not reflect the experiences of all parents. Also, our study was conducted in English, so our results do not offer insight into non-English-speaking parents.

Additionally, the data collected were limited with regard to parent demographics and psychological history. Information regarding parent gender and pre-existing mental health diagnosis would be useful for further understanding the emotional experience of the parent cohort. Previous research identifies gender as an important predictor of psychological outcomes [25]. Although we lacked information on premorbid anxiety and depression, the significantly higher prevalence of these conditions observed in our study population compared to the general population supports the robustness of our findings. This emphasises the considerable psychological burden experienced by parents and highlights the critical gap in resources and support needed to address these challenges over time.

### 4.2. Future Research Directions

One area for further investigation is exploring the sub-populations of parents who exhibit different trajectories of psychological distress. Our study contributes to the existing literature in that the majority of parents adjust well over time, but a significant number have ongoing poor emotional wellbeing [58,60,61]. Further understanding the characteristics of parents who remain distressed compared to those who demonstrate improved emotional experience could provide valuable insights into risk and resilience factors, enabling tailored interventions to better support vulnerable groups. Future research into institutional factors such as available psychological services would also lead to improved understanding and support of the psychosocial experience of parents.

Expanding on the patterns observed in this study, future research should also focus on developing interventions to address specific aspects of parental HRQoL, particularly regarding the second peak of anxiety towards the end of treatment or the sustained levels of anger observed in this population. Interventions could be assessed using the ET tool and PROMIS questionnaires which have been shown to be a valid and reliable option for monitoring parental psychological experience in this population [31]. For example, with ongoing parent recruitment through the ASSET HRQoL sub-study, we could assess how parents who initially have clinically elevated need for help scores subsequently respond following an intervention.

### 4.3. Clinical Implications

The high prevalence of anxiety, depression, anger, distress and the need for help among parents of children with ALL support the incorporation of emotional screening at diagnosis and at critical treatment milestones such as the end of treatment and transition to survivorship care [59].

There is robust evidence supporting the use of tools such as the PAT for screening in paediatric oncology to identify family psychosocial risks [29,43]. Our findings, including the correlation between the ET tool responses and more extensive PROMIS questionnaires, support the use of the ET tool for screening for impaired emotional wellbeing and documenting trends over time [62]. The ET tool could be used to evaluate the effectiveness of psychosocial interventions within an institution and serve as an adjunct to tools like the PAT to identify parents and carers experiencing heightened distress.

Recognizing the high psychological burden carried by parents reinforces the importance of a holistic, family-centred approach to paediatric oncology care. Providing resources not only for the child but also for their caregivers ensures that the emotional and social needs of the entire family unit are met, ultimately improving overall health outcomes and quality of life [21,31,63].

## 5. Conclusions

Our results highlight the significant and often enduring psychological burden experienced by parents of children with ALL. These results emphasise elevated levels of anxiety, depression, distress, anger, and the need for help compared to the general Australian and New Zealand populations. Despite a general trend toward reduced distress, anxiety and depression over time, a substantial proportion of parents continued to experience clinically significant challenges beyond two years post-diagnosis, underscoring the long-term psychological impact of caregiving.

Further research is needed to explore risk and resilience factors, develop tailored interventions addressing specific emotional needs and evaluate the impact of these interventions using validated tools like the ET tool and PROMIS questionnaires. Addressing identified gaps in parental emotional support would improve the overall wellbeing of families navigating childhood ALL treatment, thereby reducing the morbidity associated with paediatric ALL.

## Figures and Tables

**Figure 1 cancers-17-01238-f001:**
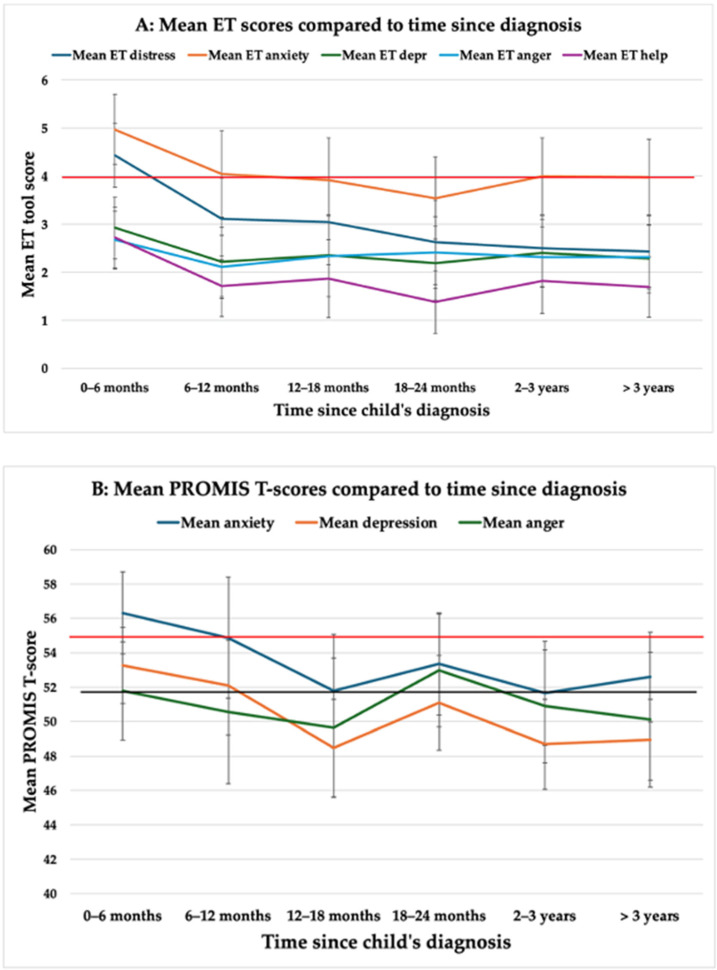
(**A**) Mean Emotion Thermometer (ET) scores compared to time since diagnosis. Red horizontal line indicates the clinical cut-off of ≥ 4. Error bars indicated 95% confidence interval. (**B**) Mean PROMIS T-scores compared to time since diagnosis. Red horizontal line indicates the clinical cut-off of 0.5 SD above the mean. Error bars indicate 95% confidence interval.

**Figure 2 cancers-17-01238-f002:**
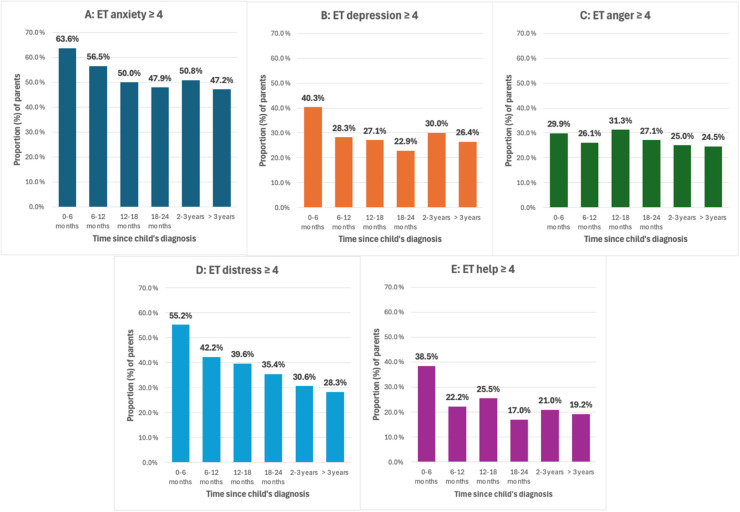
Proportions of parents reporting Emotion Thermometer (ET) scores above the clinical cut-off (≥ 4) across time since their child’s diagnosis. (**A**) ET anxiety, (**B**) ET depression, (**C**) ET anger, (**D**) ET distress and (**E**) ET need for help.

**Figure 3 cancers-17-01238-f003:**
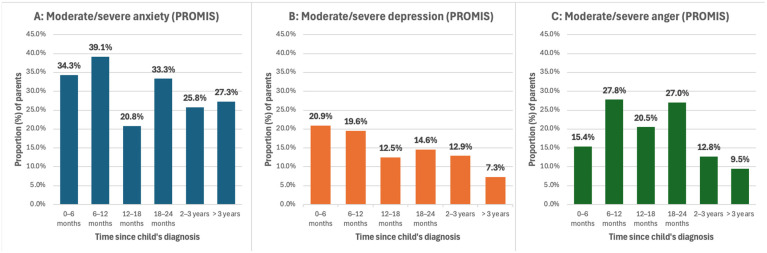
Proportion of parents reporting PROMIS questionnaire responses in the moderate to severe range. (**A**) PROMIS anxiety, (**B**) PROMIS depression and (**C**) PROMIS anger.

**Table 1 cancers-17-01238-t001:** Demographics of the patient cohort in the ASSET HRQoL sub-study (n = 117 children with ALL).

Patient Information	n	%
Median age in years at diagnosis (range)	7 (0.5–18)	
Age < 10	85	72.6
Age ≥ 10	32	27.4
Male	71	60.7
Female	46	39.3
Metropolitan *	90	76.9
Inner regional *	22	18.8
Outer regional or remote *	5	4.3
SEIFA ^#^ rank <20th centile	7	6.0
SEIFA rank 20–40th centile	19	16.2
SEIFA rank 40–60th centile	21	17.9
SEIFA rank 60–80th centile	27	23.1
SEIFA rank >80th centile	32	27.4
Unknown (New Zealand)	11	9.4
Treated on COG protocols	75	64.1
Treated on BFM protocols	42	35.9
Low/standard risk ALL	41	35.0
Medium risk ALL	35	29.9
High risk ALL	41	35.0
Presence or development of treatment-related toxicity	35	29.9
Child relapsed during study	4	3.4
Child died during study	2	1.7

* Accessibility/remoteness index (ARIA) [46]. ^#^ Socio-economic indexes for areas (SEIFA) score in Australia. The SEIFA score indicates relative socio-economic advantage and disadvantage using Census data, with a lower score indicating a higher level of socio-economic disadvantage [47].

**Table 2 cancers-17-01238-t002:** Prevalence of clinically significant distress, anxiety, depression, anger and the need for help as measured by the Emotion Thermometer (ET) and PROMIS questionnaires in the ASSET HRQoL sub-study (eight measures in total) compared to time since diagnosis.

Time Since Child’s Diagnosis	Total Number of Survey Responses	Number of Clinically Elevated Measures in Parent Responses = n (%)				
		0	1	2	3	≥4
0–6 months	67	9 (13.4)	11 (16.4)	10 (14.9)	10 (14.9)	27 (40.3)
6–12 months	46	15 (32.6)	6 (13.0)	2 (4.3)	6 (13.0)	17 (37.0)
12–18 months	48	21 (43.8)	6 (12.5)	3 (6.3)	2 (4.2)	16 (33.3)
18–24 months	48	17 (35.4)	9 (18.8)	4 (8.3)	3 (6.3)	15 (31.3)
2–3 years	63	27 (42.9)	8 (12.7)	1 (1.6)	8 (12.7)	19 (30.2)
>3 years	55	21 (38.2)	9 (16.4)	6 (10.9)	6 (10.9)	13 (23.6)
All time points	327	110 (33.6)	49 (15.0)	26 (8.0)	35 (10.7)	107 (32.7)

**Table 3 cancers-17-01238-t003:** Single variable and multiple variable mixed model analysis of PROMIS questionnaire compared to variables including time since diagnosis, child’s age at diagnosis, child’s sex, SEIFA index, ALL study group (COG or BFM), ALL risk stratification and presence of toxicity.

		Single Variable Analysis		Multiple Variable Analysis	
		Estimate (95% CI)	*p*-Value	Estimate (95% CI)	*p*-Value
ET Distress	Time since diagnosis	−0.04 (−0.06 to −0.01)	**0.001 ***	−0.07 (−0.13 to −0.02)	**0.013 ***
	Child’s age at diagnosis	0.02 (−0.08 to 0.13)	0.67	0.10 (−0.02 to 0.36)	0.49
	Child’s sex	0.40 (−0.43 to 1.23)	0.34	0.06 (−1.47 to 1.59)	0.94
	SEIFA index	−0.01 (−0.03 to 0.004)	0.13	−0.01 (−0.04 to 0.02)	0.38
	ALL study group (COG/BFM)	−0.35 (−1.19 to 0.49)	0.41	−0.02 (−1.75 to 1.35)	0.80
	Risk stratification	−0.33 (−1.35 to 0.70)	0.53	0.73 (−1.29 to 2.74)	0.48
	Presence of toxicity	0.29 (−0.65 to 1.23)	0.54	0.22 (−1.26 to 1.70)	0.77
ET Anxiety	Time since diagnosis	−0.02 (−0.05 to 0.001)	0.06	−0.06 (−0.12 to 0.01)	0.11
	Child’s age at diagnosis	−0.03 (−0.14 to 0.09)	0.63	0.04 (−0.29 to 0.38)	0.80
	Child’s sex	0.44 (−0.45 to 1.33)	0.33	0.68 (−1.21 to 2.56)	0.48
	SEIFA index	−0.01 (−0.03 to 0.004)	0.15	−0.02 (−0.05 to 0.02)	0.38
	ALL study group (COG/BFM)	0.09 (−0.91 to 0.99)	0.85	0.003 (−1.90 to 1.90)	1.00
	Risk stratification	−0.38 (−1.46 to 0.70)	0.49	−1.05 (−3.18 to 1.08)	0.33
	Presence of toxicity	0.32 (−0.68 to 1.33)	0.53	0.28 (−1.51 to 2.06)	0.76
ET Depression	Time since diagnosis	−0.01 (−0.03 to 0.01)	0.33	−0.02 (−0.06 to 0.01)	0.21
	Child’s age at diagnosis	0.02 (−0.08 to 0.12)	*0.74*	0.03 (−0.12 to 0.18)	0.69
	Child’s sex	0.40 (−0.41 to 1.21)	0.33	0.22 (−0.89 to 1.32)	0.70
	SEIFA index	−0.01 (−0.02 to 0.01)	0.42	−0.001 (−0.02 to 0.02)	0.95
	ALL study group (COG/BFM)	−0.36 (−1.18 to 0.46)	0.39	−0.04 (−1.21 to 1.12)	0.94
	Risk stratification	−0.36 (−1.34 to 0.62)	0.47	−0.39 (−1.79 to 1.01)	0.59
	Presence of toxicity	0.13 (−0.79 to 1.05)	0.78	0.50 (−0.76 to 1.76)	0.44
ET Anger	Time since diagnosis	−0.01 (−0.03 to 0.01)	0.53	−0.006 (−0.03 to 0.02)	0.62
	Child’s age at diagnosis	−0.01 (−0.11 to 0.09)	0.84	−0.05 (−0.16 to 0.07)	0.42
	Child’s sex	0.38 (−0.35 to 1.11)	0.31	0.30 (−0.47 to 1.07)	0.45
	SEIFA index	−0.02 (−0.03 to −0.004)	**0.013 ***	−0.02 (−0.03 to −0.001)	**0.038 ***
	ALL study group (COG/BFM)	−0.76 (−1.49 to −0.04)	**0.040 ***	−0.48 (−1.27 to 0.32)	0.24
	Risk stratification	0.21 (0.68 to 1.10)	0.64	0.15 (−0.79 to 1.09)	0.76
	Presence of toxicity	0.39 (−0.42 to 1.20)	0.34	0.14 (−0.73 to 1.02)	0.75
ET Need for Help	Time since diagnosis	−0.02 (−0.04 to 0.00)	**0.045 ***	−0.04 (−0.08 to 0.01)	0.11
	Child’s age at diagnosis	−0.06 (−0.16 to 0.03)	0.19	0.07 (−0.23 to 0.38)	0.63
	Child’s sex	0.090 (−0.68 to 0.86)	0.82	0.23 (−1.47 to 1.92)	0.79
	SEIFA index	−0.01 (−0.03 to 0.003)	0.11	−0.01 (−0.04 to 0.02)	0.40
	ALL study group (COG/BFM)	0.12 (−0.65 to 0.89)	0.75	0.15 (−1.54 to 1.84)	0.86
	Risk stratification	0.11 (−0.81 to 1.04)	0.81	−0.68 (−2.56 to 1.20)	0.48
	Presence of toxicity	0.10 (−0.75 to 0.95)	0.81	0.15 (−1.34 to 1.64)	0.84
PROMIS Anxiety	Time since diagnosis	−0.19 (−0.28 to −0.10)	**<0.001 ***	−0.18 (−0.28 to −0.09)	**<0.001 ***
	Child’s age at diagnosis	−0.29 (−0.75 to 0.16)	0.21	0.14 (−0.38 to 0.66)	0.60
	Child’s sex	2.83 (−0.70 to 6.36)	0.11	2.17 (−1.59 to 5.92)	0.26
	SEIFA index	0.001 (−0.07 to 0.07)	0.97	−0.002 (−0.08 to 0.07)	0.95
	ALL study group (COG/BFM)	2.20 (−1.38 to 5.78)	0.23	3.30 (−0.62 to 7.22)	0.10
	Risk stratification	0.69 (−3.62 to 4.99)	0.75	0.54 (−4.17 to 5.26)	0.82
	Presence of toxicity	−2.60 (−6.42 to 1.21)	0.18	−2.57 (−6.79 to 1.65)	0.23
PROMIS Depression	Time since diagnosis	−0.12 (−0.21 to −0.04)	**0.004 ***	−0.13 (−0.22 to −0.04)	**0.006 ***
	Child’s age at diagnosis	−0.15 (−0.58 to 0.27)	0.48	0.28 (−0.21 to 0.77)	0.26
	Child’s sex	1.48 (−1.71 to 4.67)	0.36	0.41 (−3.04 to 3.85)	0.82
	SEIFA index	−0.007 (−0.07 to 0.06)	0.82	0.003 (−0.06 to 0.07)	0.92
	ALL study group (COG/BFM)	−0.23 (−3.47 to 3.01)	0.89	0.30 (−3.29 to 3.89)	0.87
	Risk stratification	−0.85 (−4.73 to 3.04)	0.67	−1.60 (−5.85 to 2.66)	0.46
	Presence of toxicity	−2.28 (−5.56 to 1.00)	0.17	−2.53 (−6.18 to 1.13)	0.18
PROMIS Anger	Time since diagnosis	−0.024 (−0.14 to 0.09)	0.68	−0.03 (−0.16 to 0.10)	0.63
	Child’s age at diagnosis	−0.17 (−0.67 to 0.32)	0.50	−0.02 (−0.60 to 0.56)	0.94
	Child’s sex	1.64 (−2.44 to 5.72)	0.43	2.05 (−2.39 to 6.49)	0.36
	SEIFA index	0.004 (−0.07 to 0.08)	0.92	−0.02 (−0.10 to 0.07)	0.67
	ALL study group (COG/BFM)	1.38 (−2.69 to 5.45)	0.50	1.37 (−3.16 to 5.90)	0.55
	Risk stratification	0.11 (−4.69 to 4.91)	0.96	−0.89 (−6.28 to 4.51)	0.75
	Presence of toxicity	−3.28 (−7.33 to 0.78)	0.11	−4.24 (−8.81 to 0.34)	0.07

CI = confidence interval; SEIFA = Socio-Economic Indexes for Areas; ALL = acute lymphoblastic leukaemia; COG = Children’s Oncology Group; BFM = Berlin–Frankfurt–Münster; ET = Emotion Thermometer; PROMIS = Patient-Reported Outcome Measurement Information System. *: bolded results indicate statistically significant associations with *p*-value < 0.05.

**Table 4 cancers-17-01238-t004:** Correlation between anxiety, depression and anger scores when measured by the ET tool or the PROMIS questionnaires.

	Pearson Correlation (95% CI)	*p*-Value
Anxiety	0.44 (0.35–0.52)	**<0.001 ***
Depression	0.34 (0.24–0.43)	**<0.001 ***
Anger	0.17 (0.05–0.30)	**0.008 ***

* bolded results indicate statistically significant correlations with *p*-value < 0.05.

**Table 5 cancers-17-01238-t005:** Measure of agreement (Kappa statistic) between clinically elevated anxiety, depression and anger as measured by the ET tool or the PROMIS questionnaires.

	Kappa Statistic	*p*-Value
Anxiety	0.14	**<0.001 ***
Depression	0.16	**<0.001 ***
Anger	0.09	0.060

* bolded results indicate statistically significant agreement with *p*-value < 0.05.

## Data Availability

Due to privacy and ethical restrictions, these data are not publicly available.

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
