# Peer review of "Psychosocial Outcomes in Parents of Children with Acute Lymphoblastic Leukaemia in Australia and New Zealand Through and Beyond Treatment"

_cancers, 2025, doi:10.3390/cancers17071238_

Round 1
Reviewer 1 Report
Comments and Suggestions for Authors
This paper on prospective distress of parents of children with ALL in Australia and New Zealand is important because it described functioning over a longer period of time. A total of 117 parents participated who were asked to do a distress assessments every 2 months. Parents were not very responsive to this (median number of assessments is 2). Emotion thermometers were used (n=5) and PROMIS item banks (n=3).
General remarks
Although this paper is interesting, the outcomes are not very new and not very surprising, they all underscore the already existing knowledge: highest distress after diagnosis and during treatment, thereafter going down. Major concerns are the response (how many children were in the study of which the parents participated), the background of the reporter (it is not clear how many mothers or fathers participated while gender of the caregiver is one of the most important predictors) and the risk factors in the model (only medical/social demographic while we know from research psychosocial factors are most important). This should be adressed in more detail
Title:
- The wording of evaluation of distress measures is confusing- I would suggest to leave measures
Abstract:
- Aims are not clearly presented. Only describing experiences and not the risk factors
- Method does not include statistics
Introduction
- This is a good overview of already existing work in the field. While the overview show risk and protective factors such as co-caregivers, social support, social isolation- none of these psychsocial factors have been included in the study and could be added to the risk model. Little evidence is give about medical factors such as low/medium risk for distress etc while these are in the model. This should be addressed in more detail in the discussion that this is a limitation and what should be considered in the future. The PAT is an often used family risk screener which is important in this context and this should be outlined in more detail- also in the introduction with literature.
- You write psychological comorbidities: doe you refer to the outcomes here? This is confusing
- The secondary aim with risk factors: please add which you are studying
Method
- It is clearly described what help would be provided when a high score on need of help was seen. Do you have any impression whether this has influenced the scores of the parents thereafter?
- The procedure is not very clear- how were child/parents approached for this study? And how often were they called to fill in their questionnaires? Who was responsible for this- all sites themselves?
Results
- You describe opt-in HRQoL study. Please provide more detailed information on the number of families who opted-in and who did not. Potential biases in respons and representativity should be a major concern and discussion point. Because the respons of the participating families is overall also very low.
- Inclusion time is not clear- during how long parents were included for this trial?
- Table 1 provides patient information, no background information of parents is provided. In the discussion it reads information on parent gender would have been useful (line 408). It is very difficult to understand the outcomes if we do not know number of participating mothers and fathers, because we know mothers will likely score higher. This is a major concern of the study, as well as no information on marital status. Is it possible to get the information from the families about who filled in the questionnaires- that is getting the gender?
- 2 and 3.3, 3.4 provide the same data but in a different way: mean scores and percentages above the clinical range/SD. Considering the length of the paper I would recommend to change this or make supplemental tables
- 6 about the correlations. I wonder whether this adds to the manuscript. If you keep this in- it should also be a research aim because it is now a complete results paragraph
Discussion
- Several sites have participated in this study who will all have different psychosocial services. What are the difference between the sites? Can it have influenced the results? Please reflect on this potential confounder
- Line 350: it reads detailed risk factor analysis. I believe that this description should be adjusted because, in reality, only a very modest analysis has been conducted. Many studies have shown that medical variables have little explained variance in explaining psychological outcomes. The study is lacking psychological variables- or a biopsychosocial approach.
- 3 clinical implications should include the standards of screening/ PAT
- Limitations should be extended with respons. 117 parents of how many children win the study?
Minor
- Please always provide ET first and thereafter PROMIS
Reviewer 2 Report
Comments and Suggestions for Authors
Congratulations on this well written paper. I found it to be very clear and easy to follow.
My only very minor comment is what software/platform was used to collect the online survey data? (line 124)
Author Response
Comment 1: Congratulations on this well written paper. I found it to be very clear and easy to follow.
RESPONSE 1: Thank you for this review, we are very pleased to be able to present this study, and no need for further changes.
Comment 2: What software/platform was used to collect the online survey data? (line 124)
RESPONSE 2 : The REDCap software was used to collect online survey data, and we have added this to the Methods section 2.2 (lines 150-151): "Following consent, eligible parents were sent an invitation email to complete their study questionnaire via Research Electronic Data Capture (REDCap)."
Reviewer 3 Report
Comments and Suggestions for Authors
The manuscript “Prospective longitudinal evaluation of distress measures for parents of children with acute lymphoblastic leukaemia” submitted to the journal Cancers is deal with to investigation the psychological experience of Australian and New Zealand parents of children diagnosed with acute lymphoblastic leukaemia, including prevalence and time course of psychological comorbidities and identifying risk factors for increased parental distress, anxiety, depression, anger and need for help. The topic discussed is very important for increasing the long-term survivals and improving treatment of children with acute lymphoblastic leukaemia.
I would like to make some comments:
- In the item Introduction please eliminate repetitions. For example:
lines 69-71: “The distress and emotional burden on parents has been shown to negatively impact on quality of life for both parents themselves as well as children [5–7]”.
lines 73-74: “Increased parental distress can negatively impact on child wellbeing and distress levels [11].”
- Please indicate in the Materials and Methods section that all parents/guardians of children provided informed voluntary consent to participate in the study and personal data processing according to the Helsinki Declaration of the World Medical Association (WMA Declaration of Helsinki - Ethical Principles for Medical Research Involving Human Subjects, 2013).
- It is desirable to provide a questionnaires “The ET tool” and “PROMIS” in the section Supplemental materials.
- The sentences “Parents were also given the opportunity to have their concerns raised with their child’s local treating team, or the opportunity to discuss their need for help with a study psychologist via telephone conversation” should be moved from the section 2.3. Statistical Analysis in another section as it does not relate to statistical processing methods. Having this information is very important. It may be necessary to create a separate section on the assistance that was provided to parents.
- If there is a description of which parent (mother or father) answered the questions, how many complete and single-parent families participated in the study, this information should be included.
- In table 2, it is desirable to remove the last line (All time points 327 110 (33.6) 49 (15.0) 26 (8.0) 35 (10.7) 107 (32.7)), it is misleading with percentage values ​​greater than one hundred.
- In the note to Tables 2,3 and Figure 1,3 indicate the decoding of the abbreviations.
Reviewer 4 Report
Comments and Suggestions for Authors
This manuscript evaluates prospectively distress measures for parents of children with acute lymphoblastic leukaemia, on different months post-diagnosis.
It is of really interest, as it shows the main feelings and symptoms perceived by parents during the oncological treatment of their children and several months afterwards, explaining how these emotions change on time. They expose an adequate background to understand their research. Their statistical analysis is really exhaustive, and it is well described on their methods section.
However, in this statistical analysis, they state that (lines 152-154) "for parents indicating a clinically elevated need for help, they received a follow up phone call by a research psychologist and were offered information resources and information about local supportive services via e-mail". This in my opinion is really interesting and important for parents to deal with all these emotions, but I don't see that authors have discussed this intervention on their manuscript, and it could have modified some answers of the PROMs. I encourage authors to mention this on their discussion, as well as in their clinical implications/strengths and limitations.
Author Response
Thank you very much for taking the time to review this manuscript. Please find our responses below and the corresponding revisions/corrections in track changes in the re-submitted files.
- Comment 1: This manuscript evaluates prospectively distress measures for parents of children with acute lymphoblastic leukaemia, on different months post-diagnosis. It is of really interest, as it shows the main feelings and symptoms perceived by parents during the oncological treatment of their children and several months afterwards, explaining how these emotions change on time. They expose an adequate background to understand their research. Their statistical analysis is really exhaustive, and it is well described on their methods section.
RESPONSE 1: Thank you for this comment and feedback. We agree this is an important topic and we are grateful for the opportunity to resubmit this revision.
- Comment 2: However, in this statistical analysis, they state that (lines 152-154) "for parents indicating a clinically elevated need for help, they received a follow up phone call by a research psychologist and were offered information resources and information about local supportive services via e-mail". This in my opinion is really interesting and important for parents to deal with all these emotions, but I don't see that authors have discussed this intervention on their manuscript, and it could have modified some answers of the PROMs. I encourage authors to mention this on their discussion, as well as in their clinical implications/strengths and limitations.
RESPONSE 2: Thank you for this interesting suggestion with significant clinical implication. The idea that clinician and researcher responses to high "need for help" scores may influence subsequent scores is an interesting concept. We included these details in our methods to highlight the psychological safety measures incorporated into the study. Due to the study design, we were unable to statistically analyse the impact of this intervention on subsequent responses. However, this represents an important area for future research. With continued data collection through the ASSET registry, our study team may be able to analyse the cohort of parents who were approached based on high "need for help" scores and explore this further. We have now included in discussion in 4.2 future research directions (lines 552-555): “For example, with ongoing parent recruitment through the ASSET HRQoL substudy, we could assess how parents who initially have clinically elevated need for help scores subsequently respond following intervention.”
Round 2
Reviewer 1 Report
Comments and Suggestions for Authors
thanks you for addressing all comments in depth. I have no more question.